# Success of Thrombectomy in Management of Ischemic Stroke in Two Patients with SynCardia Total Artificial Heart in Bridge-to-Transplantation

**DOI:** 10.3390/bioengineering8090126

**Published:** 2021-09-19

**Authors:** Brendan Le Picault, Charles-Henri David, Pierre-Louis Alexandre, Cédric Lenoble, Philippe Bizouarn, Thierry Lepoivre, Nicolas Groleau, Bertrand Rozec, Hubert Desal, Jean-Christian Roussel, Thomas Sénage

**Affiliations:** 1Department of Cardiothoracic Surgery, Nantes University Hospital, 44093 Nantes, France; charleshenri.david@chu-nantes.fr (C.-H.D.); jeanchristian.roussel@chu-nantes.fr (J.-C.R.); thomas.senage@chu-nantes.fr (T.S.); 2Department of Neuroradiology, Nantes University Hospital, 44093 Nantes, France; pierrelouis.alexandre@chu-nantes.fr (P.-L.A.); cedric.lenoble@chu-nantes.fr (C.L.); hubert.desal@chu-nantes.fr (H.D.); 3Department of Cardiothoracic Anesthesiology, Nantes University Hospital, 44093 Nantes, France; philippe.bizouarn@chu-nantes.fr (P.B.); thierry.lepoivre@chu-nantes.fr (T.L.); nicolas.groleau@chu-nantes.fr (N.G.); bertrand.rozec@chu-nantes.fr (B.R.)

**Keywords:** Total Artificial Heart, TAH, SynCardia, ischemic stroke, thrombectomy, bridge-to-transplant

## Abstract

Introduction: Circulatory assistance from a SynCardia Total Artificial Heart (SynCardia-TAH) is a reliable bridge-to-transplant solution for patients with end-stage biventricular heart failure. Ischemic strokes affect about 10% of patients with a SynCardia-TAH. We report for the first time in the literature two successful thrombectomies to treat the acute phase of ischemic stroke in two patients treated with a SynCardia-TAH in the bridge-to-transplant (BTT). Case report: We follow two patients with circulatory support from a SynCardia-TAH in the bridge-to-transplant for terminal biventricular cardiac failure with ischemic stroke during the support period. An early in-hospital diagnosis enables the completion of a mechanical thrombectomy within the first 6 h of the onset of symptoms. There was no intracranial hemorrhagic complication during or after the procedure and the patients fully recovered from neurological deficits, allowing a successful heart transplant. Conclusion: This case report describes the possibility of treating ischemic strokes under a SynCardia-TAH by mechanical thrombectomy following the same recommendations as for the general population with excellent results and without any hemorrhagic complication during or after the procedure.

## 1. Introduction

Mechanical circulatory support (MCS), a left ventricular assistance device (LVAD) or a Total Artificial Heart as a bridge to a heart transplant enables patients with mono- or biventricular failure to successfully reach heart transplantation in 50 to 80% of cases [1,2]. Among this population, the outcomes and long-term survival are comparable to patients with a direct heart transplant.

Neurological complications represent one of the main complications under assistance. To avoid compromising the bridge to transplantation, both the prevention and management of strokes are critical. According to the literature, ischemic strokes occur in approximately 13% [3] to 20% of patients with an LVAD and in 8–11% [4,5] of patients assisted by a SynCardia-TAH.

The anticoagulation and platelet anti-aggregation protocol for SynCardia-assisted patients consists—according to the manufacturer and studies cited by the manufacturer—of therapeutic anticoagulation with unfractionated heparin combined with platelet mono-anti-aggregation introduced when the risk of bleeding is ruled out postoperatively after SynCardia implantation. In some cases, depending on the patient and the benefit-risk balance, dual platelet aggregation may be used in addition to therapeutic anticoagulation.

At a distance from the surgery and once the clinical situation has stabilized and the risk of bleeding has been ruled out, a relay of unfractionated heparin by an anti-vitamin K treatment can be carried out.

There are few data on the management of ischemic strokes in patients under circulatory assistance with a SynCardia-TAH (which is a Total Artificial Heart consisting of two orthotopic ventricles, four mechanical valve prostheses and a blood-machine interface represented by a polyurethane membrane mobilized by pneumatic energy).

The aim of this case report is to share a monocentric experience of the management of two patients with ischemic stroke during circulatory assistance by a SynCardia-TAH in the bridge-to-transplant. For these two patients, assistance with a SynCardia-TAH was motivated by non-reversible biventricular heart failure suitable for heart transplant with identified temporary contraindications that prohibited an immediate heart transplant.

1Patient 1 (Figure 1)

The first patient was a 44-year-old man, 1.90 m tall and weighing 94 kg (BMI 26 kg/m^2^), with end-stage ischemic heart failure, complicated by renal failure requiring extra-renal dialysis treatment. The patient was implanted with a 70cc SynCardia-TAH (body surface area: 2.22 m^2^). The anticoagulation protocol consisted of treatment with unfractionated heparin with an anti-Xa target activity of 0.20–0.40 IU/mL, associated with acetylsalicylic acid at 160 mg per day (acetylsalicylic acid was added after SynCardia implantation as soon as the hemorrhagic risk had been eliminated). He did not have any hemorrhagic complication but presented numerous thrombotic complications unrelated to heparin-induced thrombocytopenia or heparin resistance: deep vein thrombosis of the right internal jugular vein and vena cava, several small pulmonary embolisms responsible for a pulmonary infarction, right and left kidneys infarction, splenic infarction and several small ischemic strokes with no significant clinical translation (which, however, justified the administration of 160 mg of acetylsalicylic acid per day). The main thrombotic complication was the occurrence of an embolic ischemic stroke on the 57th day of assistance. This occurred due to an occlusion of the left middle cerebral artery responsible for right hemiplegia. In the 12 days prior to the stroke, his anti-Xa activity ranged from 0.3 to 0.7 IU/mL. At this time, relay by anti-vitamin K treatment was not initiated because unfractionated heparin was easier to manage in this patient’s context. The early in-hospital diagnosis of the neurological deficit allowed for rapid management by thrombo-aspiration with an aspiration catheter alone, performed 100 min after the onset of symptoms. This thrombectomy enabled immediate revascularization of the territory of the left middle cerebral artery. No hemorrhagic complications occurred following the thrombectomy despite the maintenance of anticoagulant and platelet anti-aggregation treatments without any modification in the immediate pre- and post-interventional period. Subsequently, the neurological deficit fully regressed and the patient had no adverse event. After this complication, therapeutic anticoagulation and platelet anti-aggregation with acetylsalicylic acid were continued according to the same protocol.

On day 8 after the stroke, the patient was successfully transplanted after a total of 65 days of assistance. The patient was discharged from hospital without any neurological sequelae.

2Patient 2 (Figure 2)

The second patient was a 35-year-old man, 1.75 m tall and weighing 53 kg (BMI 17 kg/m^2^), with end-stage heart failure related to dilated cardiomyopathy, secondary to anthracycline treatment for nephroblastoma in childhood, associated with non-compaction of the left ventricle. This heart failure was complicated by renal failure requiring extra-renal dialysis treatment. The patient was implanted with a 50cc SynCardia-TAH (body surface area: 1.64 m^2^). The anticoagulation protocol consisted of treatment with unfractionated heparin with a target anti-Xa activity of 0.20–0.40 IU/mL associated with acetylsalicylic acid of 75 mg per day (acetylsalicylic acid was added after SynCardia implantation as soon as the hemorrhagic risk had been eliminated). The post-operative follow-up was marked by one repeat operation for hemostasis in the first 24 h, followed by several thrombotic complications such as a pulmonary embolism, pulmonary infarction, and deep venous thrombosis of the inferior vena cava and the left iliac vein. These thrombotic complications were unrelated to heparin-induced thrombocytopenia and were attributed to both a biological inflammatory syndrome and circulatory assistance by the SynCardia. The complications led us to upgrade the anti-Xa activity target to 0.35–0.50 IU/mL after 39 days of assistance (continuing the treatment with acetylsalicylic acid as originally defined).

Despite these adjustments, the patient suffered an embolic stroke on the 56th day of assistance related to an acute occlusion of the bifurcation of the right proximal middle cerebral artery responsible for left hemiplegia. In the month prior to the stroke, the anti-Xa activity was between 0.30 and 0.55 IU/mL. At this time, relay by anti-vitamin K treatment was not initiated because unfractionated heparin was easier to manage in this patient’s context. The early in-hospital diagnosis of the neurological deficit allowed treatment by mechanical thrombectomy with a combined stent retriever and aspiration catheter technique 70 min from the onset of symptoms with no bleeding complication. Therapeutic anticoagulation and platelet anti-aggregation were continued without any modification in the immediate pre- and post-interventional periods. Revascularization of the right middle cerebral artery allowed the neurological deficit to immediately be corrected. Subsequently, the patient made a complete recovery. After this episode, therapeutic anticoagulation and platelet anti-aggregation with acetylsalicylic acid were continued according to the same protocol.

On day 5 after the stroke, the patient received a heart transplant after a total circulatory support period of 61 days. The patient was discharged from hospital with no neurological sequelae.

## 2. Discussion

So far not described in the literature, these two case reports confirm that it is possible to propose an optimal therapeutic management of ischemic strokes under a SynCardia-TAH.

The acute management of ischemic stroke in the general population has undergone several developments to improve the prognosis of this pathology. In 2015, several studies [6,7] led to a grade I recommendation to perform mechanical thrombectomy in all patients eligible for the technique during the first 6 h following the onset of symptoms.

More recently, in 2018, the delay of access to mechanical thrombectomy in selected patients was validated for up to 24 h after the onset of symptoms [8].

The technique of mechanical thrombectomy for the acute management of ischemic stroke consists of self-expanding retriever stents and/or aspiration catheters, which are introduced into the arterial network via peripheral percutaneous access and mounted, under-scopic control, via micro-guides to the site of occlusion of the cerebral artery. Stent retrievers are then deployed on the thrombus responsible for the occlusion, allowing its inclusion in the stent structure. They are then withdrawn in the deployed position, in order to extract the thrombus while avoiding its fragmentation. This technique can be associated with thrombo-aspiration by aspiration catheter. Sometimes, thrombo-aspiration alone is sufficient. One of the main risks after thrombectomy is hemorrhagic transformation in the revascularized brain parenchyma. When it occurs, this complication can be serious and lead to irreversible neurological deficits or even death.

This case report shows that it is possible to apply a similar management to patients undergoing total artificial heart support. In view of the post-operative context, intravenous thrombolysis is most often contraindicated. For these patients with an increased risk of bleeding compared to the general population (therapeutic anticoagulation and platelet anti-aggregation imposed by SynCardia-TAH assistance), thrombectomy appears to be a potentially risky option due to the higher risk of intra-or post-procedure intra-cranial hemorrhage. However, for both patients, thanks to an early clinical and scanner diagnosis, mechanical thrombectomy could be performed within 6 h of the stroke. This allowed for an ad integrum recovery of neurological deficits and neither of the patients presented intracranial hemorrhagic complications during or after the procedure. Despite the need for therapeutic anticoagulation and platelet anti-aggregation treatment to prevent thromboembolic risk under a SynCardia-TAH, the mechanical thrombectomy could be performed with efficiency and without any adverse event. Subsequently, both patients benefited from a heart transplant with good results.

## 3. Conclusions

Stroke remains a dreaded complication of all types of short- to long-term MCS, with a higher incidence in LVAD therapy than with a Total Artificial Heart.

Although patients treated with a SynCardia-TAH are not comparable to ischemic stroke patients in the general population, particularly in terms of the thrombotic and hemorrhagic risk balance, these patients’ experience show that the use of mechanical thrombectomy, when indicated, is possible with excellent results and without any hemorrhagic complication.

Thus, guidelines for the acute management of ischemic stroke appear to be applicable to patients with a SynCardia-TAH. Mechanical thrombectomy should be considered in the case of ischemic stroke under a SynCardia-TAH.

## Figures and Tables

**Figure 1 bioengineering-08-00126-f001:**
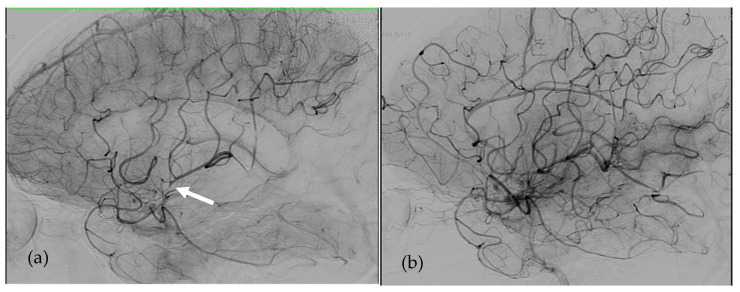
Angiographies of Left Internal Carotid Artery in patient 1. (**a**)Left Middle Cerebral Artery occlusion responsible for right hemiplegia just before revascularization (the white arrow shows the thrombus). (**b**) Early after endovascular revascularization by mechanical thrombectomy 100 min after the onset of the symptoms.

**Figure 2 bioengineering-08-00126-f002:**
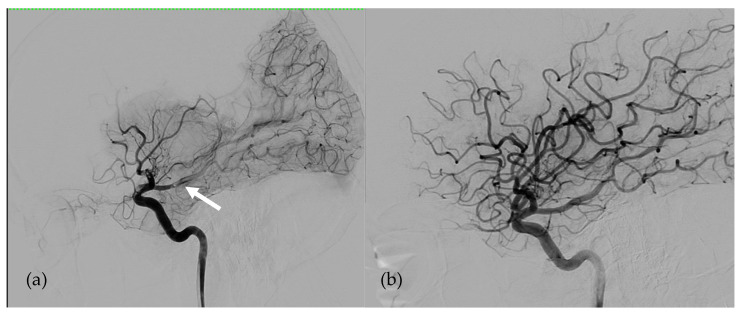
Angiographies of Right Internal Carotid Artery in patient 2. (**a**) Right Middle Cerebral Artery occlusion responsible for left hemiplegia just before revascularization (the white arrow shows the thrombus). (**b**)Early after endovascular revascularization by mechanical thrombectomy 70 min after the onset of the symptoms.

## Data Availability

Data available on request due to restrictions eg privacy or ethical.

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
