# Peer review of "Success of Thrombectomy in Management of Ischemic Stroke in Two Patients with SynCardia Total Artificial Heart in Bridge-to-Transplantation"

_bioengineering, 2021, doi:10.3390/bioengineering8090126_

Round 1

Reviewer 1 Report

an important topic in a specialized population.  The experience of this group will be of interest to others in the mechanical cardiac support community.

The author state that the anti-coagulation was restarted after the emoblectomy/thrombectomy.  How soon afterwards?

Author Response

First of all, thank you for reviewed our work.

Based on the assumption that the patients had a serious thromboembolic complication under SynCardia circulatory support (stroke), we decided not to withhold anticoagulant and antiplatelet therapy during and early after the thrombectomy procedure in order to avoid new stroke locations or further aggravation of the ongoing stroke. 

Reviewer 2 Report

I read with interest the manuscript entitled “Success of thrombectomy in management of ischemic stroke in 2 patients with SynCardia Total Artificial Heart in bridge-to-transplantation” by Le Picault et al. 

This article is well written, but the cases described are a bit long. Can you make an effort to synthesize and shorten the sentences? For example in the introduction section, the first sentence is too long (> 4 lines). 
The subject is interesting but I would like the authors to explain why a thrombectomy could not take place in such patients? why some would be reluctant? is it for technical problems? for other reasons?

Line 61 : please delete the space in “treatment”
Lines 63-65 and 100-102: why a change in typeface
Line 71: please delete “versus 75 mg for patient 2” 
Lines 75-77 and 114-115: please delete the biological parameters,  it does not provide important information for the understanding of the paper and for the management of the patient
In the case report 1, there are errors in the dates:  “embolic ischemic stroke (57th day of assistance)”, “5 days after the stoke: heart transplantation”, “transplanted after 65 days of assistance” ???

Case report 1: 
Lines 78-81 and 118-119: thrombectomy procedure isn’t clear in the two cases 
Are the devices used stent retriever devices with or not balloon guide catheters for flow occlusion, direct aspiration catheters, or combined stent retrievers/aspiration catheters ?

Author Response

First of all, thanf you for reviwed our work.

We have tried to work on the fluidity of the text and the turns of phrases by deleting certain non-essential passages. We also called on a native English speaker to correct our article.

We have also tried to answer your question about the difficulties encountered when talking about thrombectomy in patients on circulatory support. In these patients, the difficulty and risk lies in the greater risk of bleeding than in the general population. This increased haemorrhagic risk is inherent to the presence of circulatory assistance which imposes anticoagulant and antiplatelet treatments to prevent the thrombotic risk.
These patients are therefore permanently in a balance between the haemorrhagic risk and the thrombotic risk.
Even in the general population, thrombectomy can be complicated by intracranial haemorrhage. Therefore, it appears that thrombectomy may be risky in patients on SynCardia. There is little or no data in the literature on the treatment of stroke in patients on SynCardia. With this case report, we wanted to share our experience to show that thrombectomy is possible even in these patients at risk of bleeding, without being synonymous with hemorrhagic stroke complications. In order to make this clear, we have modified part of the discussion to include these elements.

Finally, we have specified the thrombectomy technique used for each patient.

Reviewer 3 Report

The authors presented two cases of mechanical thrombectomy for embolic stroke during TAH (Syncardia) support. This experience is surely very important, because supports the idea that promptly brain revascularization is mandatory for improve the prognosis of ischemic stroke. This concept is true in the general population, but it is more relevant in this kind of patients who are young and supported as a bridge to transplant. I have only minor comments:

1) can the authors better clarify the anticoagulation regimen of patients supported with TAH? are these patients treated with oral anticoagulants (Warfarin?)

2) after mechanical thrombectomy, patients cannot completely stop anticoagulation due to the mechanical support. This fact could teoretically improve the risk of hemorragic infarction of brain ischemic area. Which strategy are suggested by the authors to manage this issue?

3) TAH is a fantastic option to support end-stage HF patients, but its implant is surely more invasive and troublesome compared to a traditional LVAD. However TAH can be the only option in case of biventricular failure, with severe pulmonary hypertension. Which is the protocol at the authors center for implanting TAH versus LVAD?

Author Response

First of all, thank you for reviewed our work.

The two patients described were not being treated with avk at the time of the strokes because they were hospitalised and their discharge was not planned. In these circumstances, we felt that therapeutic anticoagulation with unfractionated heparin was more manageable in the event of the need for invasive examinations or further surgery.

Secondly, considering that the two patients had thrombotic complications of their assistants with these strokes, we had made the decision to never suspend curative anticoagulation and antiplatelet aggregation even around the thrombectomy procedure in order to avoid the increase of the stroke or the occurrence of new strokes. We would have taken the decision to suspend these treatments in the event of the occurrence of bleeding complications.

Finally, to answer your 3rd question, we have added at the end of the introduction the elements that motivated us to choose Syncardia for these two patients rather than assistance by LVAD (the main one being the presence of a non-reversible bi-ventricular failure).

Round 2

Reviewer 2 Report

I would say that the questions raised are answered quite fairly.

The authors have improved their manuscript and in particular the authors explain the place of thrombectomy in such patients.

However, the conclusion isn’t appropriate to the cases of the study. Please delete, in the last sentence, “within the first 6 hours and even 24 hours after the onset of symtoms”. Indeed, in the two cases, the thrombectomy was performed 77 and 100 minutes following the onset of symtoms.

Author Response

We agree with your request for correction on this point.

Yours sincerly.